# Can You Hear Me Now? A Benchmark for Long-Range Graph Propagation

## Abstract

Effectively capturing long-range interactions remains a fundamental yet unresolved challenge in graph neural network (GNN) research, critical for applications across diverse fields of science. To systematically address this, we introduce ECHO (Evaluating Communication over long HOps), a novel benchmark specifically designed to rigorously assess the capabilities of GNNs in handling very long-range graph propagation. ECHO includes three synthetic graph tasks – single-source shortest paths, node eccentricity, and graph diameter – each constructed over diverse and structurally challenging topologies intentionally designed to introduce significant information bottlenecks. ECHO also includes a real-world dataset, ECHO-Chem, grounded on a novel chemically-grounded application involving the prediction of atomic partial charges in molecules, which critically depends on the ability to capture intricate long-range molecular interactions. We provide an extensive benchmarking on popular GNN architectures which reveals clear performance gaps, emphasizing the difficulty of true long-range propagation and highlighting models and design choices capable of overcoming inherent limitations. ECHO thereby sets a new standard for evaluating long-range information propagation, also providing a compelling example for its need in AI for science.

## 1 Introduction

Graphs are fundamental data structures used extensively to represent complex interconnected systems, ranging from social networks and biological pathways, to communication infrastructures and molecular structures. Graph Neural Networks (GNNs) [73, 32, 67, 10, 17] have emerged as a successful methodology within deep learning, whose research community was initially driven by the development of diverse architectures capable of capturing intricate relational patterns inherent to graph-structured data, as well as impactful applications across various domains [41, 18, 36, 33, 48].

More recently, the research community has shifted its focus towards understanding and overcoming fundamental limitations of the message-passing paradigm underlying GNNs. This shift has been driven by the observation that effectively propagating information over long distances in graphs remains a significant challenge. Such challenges have been formally linked to phenomena like over-smoothing [12, 63, 65], over-squashing [2, 19], and more generally, vanishing gradients [3], all of which hinder GNN performance in tasks that require capturing long-range dependencies.

Currently, we are in the stage in which such pioneer theoretical studies need consolidation, while looking into methodological advancements that can surpass or mitigate such shortcomings. A key enabler of this progress is the establishment of solid and challenging benchmarks that can accurately assess and validate long-range propagation capacities. The availability of controlled synthetic benchmarks, should be complemented by the introduction of compelling application-driven datasets which can clearly demonstrate the practical advantages of addressing long-range propagation issues.

Submitted to 39th Conference on Neural Information Processing Systems (NeurIPS 2025). Do not distribute.

Long-range propagation capacities, in this sense, have been noted to be central in key areas of science, such as in biology [42, 25], biochemistry [38], and climate [52].

Existing graph benchmarks have, instead, focused primarily on short to medium-range tasks [8, 68, 81, 74, 78, 45, 24], often overlooking the unique challenges associated with distant information propagation. More recently, the growing interest in this challenge has motivated the community to develop a few benchmarks specifically designed to evaluate information propagation in GNNs. These include the Long-Range Graph Benchmark (LRGB) [25] and the Graph Property Prediction (GPP) dataset [34]. While this is a significant step forward compared to earlier benchmarks, it does not fully account for the need to capture the true long-range dependencies present in some real-world applications. This is due to limited size of the graphs, the absence of well-defined conditions on the expected propagation range, and the focus of the benchmarks, which is often more aimed at specific issues of over-smoothing and over-squashing, rather than providing a broader evaluation of long-range propagation capabilities. Moreover, LRGB and GPP tasks are facing a natural performance saturation, as novel methodologies are being developed and optimized on them.

Motivated by this, we introduce ECHO (Evaluating Communication over long HOps), a new benchmark designed to assess the capabilities of GNNs to exploit long-range interactions. ECHO consists of three synthetic tasks and one real-world chemically grounded task. The former are designed to provide a controlled setting to assess propagation capabilities. They comprise the prediction of shortest-path-based graph properties (i.e., node eccentricity, single-source shortest paths, and graph diameter) across a diverse graph topologies. These have been defined to increase the difficulty of effective long-range communication, as they present structural bottlenecks for the information flow. The main characteristic of these tasks is that GNNs must heavily rely on global information and effectively learn to traverse the entire graph, similarly to classical algorithms like Bellman-Ford [7]. The real-world task targets the prediction of long-range charge redistribution in molecules, a critical and practically relevant challenge in computational chemistry [22], as it underlies many fundamental processes such as chemical reactivity, molecular stability, and intermolecular interactions. Accurate modeling of these effects is essential for drug design, materials science, and biology understanding.

Our contributions can be summarized as follows:

- We introduce ECHO, a novel benchmark featuring four new tasks specifically designed to evaluate the ability of GNNs to effectively handle long-range communication in both synthetic and real-world settings. ECHO includes three synthetic tasks (collectively referred to as ECHO-Synth) with a total of 10,080 graphs, and one real-world task (ECHO-Chem) comprising 196,545 graphs, where the required propagation ranges from 17 to 40 hops.

- We propose ECHO-Chem, the first task that targets long-range interactions at atomic level for the prediction of long-range charge redistribution in molecular graphs. This makes ECHO-Chem not only a valuable task for benchmarking long-range propagation in GNNs, but also for advancing computational chemistry, where accurately modeling such interactions is notoriously challenging and computationally demanding, as highlighted by the $\approx$ **3 weeks** of computational time on our hardware configuration to produce the benchmark.

- We present a detailed analysis to demonstrate that the tasks in ECHO genuinely capture long-range dependencies, providing a rigorous evaluation of GNNs' ability to propagate information over extended graph distances.

- We conduct extensive experiments to establish strong baselines for each task in ECHO, providing a comprehensive reference point for future research on long-range graph propagation.

We openly release data at https://huggingface.co/datasets/gmander44/ECHO/tree/main and the code at https://anonymous.4open.science/r/ECHO-benchmarks

## 2   On the need of a new benchmark

We now elaborate on the need for novel benchmarks specialized on the evaluation of long-range propagation, in relation to existing datasets.

The most widely used benchmark for assessing these capabilities is arguably LRGB [25]. Its introduction in 2022 has certainly marked an important milestone and promoted the development of the field. However, despite initial rapid improvements, performance on LRGB has now plateaued, showing a noticeable deceleration in progress across the last year, as discussed in Appendix A.

In addition to this, it has to be noted that recent works [76, 5] questions the long-range nature of several LRGB tasks, revealing that a subset of tasks is inherently local, rather than requiring long-range diffusion, and that the benchmark itself is highly sensitive to hyperparameter tuning. Other benchmarks propose synthetic tasks on generated structures, including the Tree-Neighborhood [2], Graph Property Prediction [34], graph transfer [19, 35], GLoRA [85], and Barbell and Clique graphs [4]. Indeed, most of these tasks are originally designed to address narrow challenges that prevent long-range propagation, such as over-smoothing [12, 63, 65] and over-squashing [2, 19]. These phenomena, while related, do not necessarily capture the full spectrum of challenges associated with long-range communication. Moreover, despite being designed to test the ability of GNNs to overcome these limitations, these datasets typically involve small graphs with limited-size diameters. This inherently restricts the propagation radius, creating a significant gap between the benchmark tasks and real-world problems that require much deeper propagation across significantly larger structures.

The limitations highlighted above suggest the need for a new benchmark that reflects the challenges and opportunities in long-range GNN research. An effective benchmark should provide tasks that explicitly test a model's capacity to traverse extensive graph structures, effectively aggregate global information, and adapt to diverse topological constraints. Moreover, as the field has matured and a wide range of models have been established, ranging from graph transformers [70, 64] to multi-hop GNNs [1, 39] and others [69], it seems timely to introduce a new benchmark that can accurately assess the long-range propagation skills of these families of models, now that they are well understood and consolidated.

ECHO addresses this scenario by a suite of synthetic and real-world tasks with clearly defined long-range propagation needs, providing a clear target for the evaluation of this property. Specifically, ECHO tasks require computing either shortest paths between all nodes or long-range charge redistribution, with clearly defined propagation ranges between 17 and 40 hops, depending on the specific graph structure. This explicit range ensures that models failing to capture dependencies within this span are underreaching and have poor long-range capabilities.

The ECHO-Chem molecular task has strong value per-se. It proposes a novel, practical and high-impact challenge for learning models in computational chemistry [22]. Previous popular benchmark in this domain [74, 78, 45, 81, 25] focused on the prediction of molecular-level properties, such as solubility or HIV inhibition, which are short-range tasks. This is evident when they can be reduced to the problem of counting small-dimensional local substructures (ie with lenght smaller than 7) [9]. Differently, ECHO-Chem is the first graph benchmark that targets long-range interactions at the atomic level, i.e., the microscopic scale. ECHO-Chem task is not only inherently long-range, but also particularly challenging as it requires accurate modeling of charge distributions and of the complex atomic interactions. This makes it a computationally expensive task to be solved with current computational chemistry tools. We provide further details on computational complexity of the quantum simulations in Appendix G.

Therefore, ECHO-Chem sets a new standard for evaluating long-range graph information propagation, as well as it provides a compelling application of AI for science and chemistry, enabling faster predictions with potential impact on drug/material design or understanding biological functions.

## 3  The ECHO Benchmark

In this section, we introduce a suite of datasets designed to rigorously evaluate the long-range information propagation capabilities of GNNs. Our benchmark consists of two complementary components: a set of algorithmically constructed tasks and a chemically grounded real-world dataset.

The synthetic component includes classical graph-theoretic problems—single-source shortest path, node eccentricity, and graph diameter—posed across diverse graph topologies designed to induce structural bottlenecks and challenge multi-hop message passing. These tasks isolate long-range dependencies and enable controlled analysis of model behavior under varying topological conditions.

The chemical benchmark targets a practically relevant and physically grounded task in computational chemistry: predicting long-range charge redistribution in molecules. This problem, rooted in electronic structure modeling, reflects realistic charge transfer phenomena and builds upon prior work in quantum-accurate deep learning models for molecular systems [51, 84].

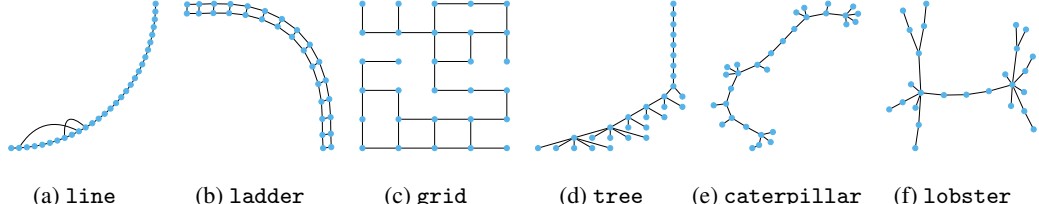

|     (a) `line`     |     (b) `ladder`     |     (c) `grid`     |     (d) `tree`     |     (e) `caterpillar`     |     (f) `lobster`     |

Figure 1: Visualization of the proposed topologies in the synthetic dataset. In all graphs, $N = 30$

### 3.1 The `ECHO-Synth` dataset

The algorithmic dataset is designed to benchmark GNNs on tasks that require long-range information propagation across a diverse set of graph topologies. It focuses on three graph property prediction tasks: **Single Source Shortest Path** (`sssp`), **Node Eccentricity** (`ecc`), and **Graph Diameter** (`diam`). Among these, `sssp` and `ecc` are node-level tasks requiring the prediction of a scalar value per node, while `diam` is a graph-level task requiring a single prediction for the entire graph. We refer to this dataset as `ECHO-Synth`.

These tasks were intentionally selected due to their heavy reliance on global information. For example, solving `sssp` from a given source node requires identifying shortest paths to all other nodes [20]—information that often spans the entire graph. Eccentricity builds on this by requiring the longest shortest path from each node, demanding complete graph awareness. Diameter is even more global, involving the longest shortest path between *any* two nodes [16]. Classical algorithms like Dijkstra's [20] and Bellman-Ford [7], which perform complete graph traversal, illustrate the challenge these tasks pose for GNNs, which rely on localized message passing. To prevent models from relying on input features rather than learning structural patterns, each node is assigned a uniformly distributed random scalar feature $r \sim \mathcal{U}(0, 1)$. Additionally, for the `sssp` task, a binary indicator is included to mark the source node. This ensures that the model can distinguish the source while maintaining uniform input statistics across tasks.

Table 1: Statistics of the proposed dataset.

| Dataset | # Graphs | Avg Nodes | Avg Deg. | Avg Edges | Avg Diam | # Node Feat | # Edge Feat | # Tasks |
|---|---|---|---|---|---|---|---|---|
| `ECHO-Synth` | 10,080 | $83.69_{\pm 66.24}$ | $2.53_{\pm 1.19}$ | $211.63_{\pm 209.39}$ | $28.50_{\pm 6.92}$ | 2 | None | 3 |
| `line` | 1,680 | $75.60_{\pm 27.32}$ | $2.37_{\pm 0.10}$ | $90.10_{\pm 33.89}$ | $28.50_{\pm 6.92}$ | 2 | None | 3 |
| `ladder` | 1,680 | $56.52_{\pm 13.82}$ | $2.92_{\pm 0.02}$ | $82.54_{\pm 20.72}$ | $28.50_{\pm 6.92}$ | 2 | None | 3 |
| `grid` | 1,680 | $193.10_{\pm 93.10}$ | $2.95_{\pm 0.12}$ | $288.32_{\pm 145.29}$ | $28.50_{\pm 6.92}$ | 2 | None | 3 |
| Tree | 1,680 | $60.42_{\pm 17.17}$ | $1.96_{\pm 0.01}$ | $59.42_{\pm 17.17}$ | $28.50_{\pm 6.92}$ | 2 | None | 3 |
| `caterpillar` | 1,680 | $34.71_{\pm 7.96}$ | $1.94_{\pm 0.02}$ | $33.71_{\pm 7.96}$ | $28.50_{\pm 6.92}$ | 2 | None | 3 |
| `lobster` | 1,680 | $81.79_{\pm 25.46}$ | $1.97_{\pm 0.01}$ | $80.79_{\pm 25.46}$ | $28.50_{\pm 6.92}$ | 2 | None | 3 |
| `ECHO-Chem` | 196,545 | $73.74_{\pm 13.23}$ | $2.09_{\pm 0.04}$ | $76.92_{\pm 13.29}$ | $23.66_{\pm 2.66}$ | 2 | 2 | 1 |

**Dataset Construction.** This dataset includes six distinct families of graph topologies i.e., line, ladder, grid, tree, caterpillar, and lobster (see Figure 1), each selected to highlight different structural and propagation characteristics. The `line` graph (Figure 1 (a)) serves as a simple but non-trivial baseline. To introduce non-local interactions, we modify it with stochastic skip connections: each node has a 20% chance of forming an edge to another node 2–6 hops away. Building on this, the `ladder` topology (Figure 1 (b)) consists of two parallel `line` graphs connected by one-to-one cross-links, enabling richer routing possibilities and redundancy in message pathways. The `grid` topology (Figure 1 (c)) represents a 2D lattice structure where edges are independently removed with a 20% probability. This results in irregular neighborhoods and broken spatial symmetries.

To model scale-free and hierarchical structures, we include `tree`-structured graphs (Figure 1 (d)) generated through preferential attachment. A new node connects to an existing one with probability proportional to $k_i^{\alpha}$, where $k_i$ represents the degree of the $i$-th node (with $\alpha = 3$), leading to the formation of high-degree hubs and reflecting connectivity patterns often seen in natural networks. The `caterpillar` topology (Figure 1 (e)) augments a central linear backbone with peripheral nodes attached randomly along the spine, combining features of chain-like and tree-like graphs to create

moderate branching and directional flow. Extending this idea, the `lobster` graph (Figure 1 (f)) adds a third hierarchical layer: nodes in the outermost layer connect only to intermediate nodes, resulting in deeper branching while preserving an overall elongated structure. This configuration is especially useful for testing the limits of multi-hop message passing under structured constraints.

Beyond their long-range dependencies, the complexity of the synthetic tasks is further increased by the presence of **topological bottlenecks**, which pose significant challenges to GNN based on message passing [29]. Bottlenecks emerge in graphs where information flow between distant nodes is constrained to pass through a small subset of intermediary nodes, thereby restricting the bandwidth of information flow. This structural constraint can increse the risk of *over-squashing*, a phenomenon in which exponentially growing information is aggregated into the low-dimensional node representations [2]. As a result, critical signals may be compressed or lost during propagation, severely limiting the model's capacity to distinguish and preserve meaningful long-range interactions [77, 19].

Graph families in synthetic dataset are explicitly designed to expose models to such bottlenecks. For example, in the `line` topology information between distant nodes must propagate sequentially through a single path, making each node along the path a critical bottleneck. Similarly, `tree`-structured graphs inherently introduce bottlenecks at branch points and hierarchical layers, where entire subtrees depend on narrow pathways for communication with the rest of the graph. The `caterpillar` and `lobster` graphs further reinforce this pattern by adding additional peripheral layers while maintaining centralized backbones, exacerbating the bottleneck effect in their hierarchical layouts. Even in the more uniform `grid` topology, bottlenecks are implicitly introduced through random edge deletions, which can disrupt regular pathways and force information to traverse suboptimal and congested routes.

**Dataset Split.** To support robust evaluation, we generate graphs with target diameters in the range $d \in [17, 40]$, capturing diverse long-range interaction scenarios. For each of the six graph topologies and each diameter value, we produce 70 unique graphs, yielding a total of $70 \times 24 \times 6 = 10,080$ graphs. To ensure consistent and unbiased evaluation, we partition these graphs into training, validation, and test splits in a stratified manner. Specifically, for each topology and diameter combination, we assign 40 graphs to the training set, 15 to the validation set, and 15 to the test set. This strategy guarantees that all splits share the same distribution over both graph topologies and diameter values, which are uniformly sampled. Consequently, models are evaluated on data that is statistically aligned with the training set, avoiding distributional shifts and ensuring fair comparison across methods. Detailed dataset statistics are reported in Table 1 and Appendix E.

### 3.2 `ECHO-Chem` **Dataset**

Molecular property prediction is a cornerstone application of GNNs, with common benchmarks involving graph-level prediction tasks such as molecular fingerprint [23], solubility, toxicity and various chemical properties [15, 46]. One fundamental task in this domain is the prediction of atomic partial charges—continuous, atom-level properties that reflect the electron distribution within a molecule. Accurate charge prediction is essential for modeling molecular interactions, reactivity, and electrostatic behavior. Figure 2 illustrates this task on the 3D molecular graph of caffeine, where each atom is colored according to its predicted partial charge.

Traditionally, partial charges are computed using quantum mechanical methods, especially Density Functional Theory (DFT) or other quantum chemical simulations. While these methods provide high accuracy, their computational cost—arising from solving complex equations—limits their scalability to large molecular datasets or high-throughput tasks. Specifically, high-accuracy simulations require several minutes to process a single molecule. We report a quantitative description of DFT and non-DFT quantum simulation efficiency in Appendix G.

A significant challenge for Machine Learning (ML) methods addressing partial charge prediction is effectively capturing long-range dependencies across molecular graphs. Specifically, here we will refer as "long-range" in the graph space, (e.g., node separated by many hops), rather than purely spatial distance. The three-dimensional configuration of molecules greatly intensifies this task complexity, as distant atoms in the graph topology can still exert significant influence on atomic electronic properties. Such non trivial, long-range interdependencies become increasingly challenging to model accurately as molecular graph diameter grow. To systematically address this challenge, we introduce `ECHO-Chem`, with the specific aim to stress long-range dependencies in a real-world

scenario. `ECHO-Chem` task is formulated as a node-level regression problem: for each molecular graph, the model must predict the partial charge of every atom.

Beyond serving as a rigorous benchmark for GNN architectures, this dataset has strong potential for practical impact in terms of ML application in science and chemistry. Capturing these sophisticated long-range interactions can significantly improve efficiency of predicting atomic partial charges, and potentially serving as accurate and computational inexpensive initialization for subsequent quantum mechanical simulations. Such improvements could substantially accelerate computational chemistry workflows, facilitating rapid exploration of the large molecular space.

**Dataset Construction.** Comprising approximately $200,000$ molecular graphs selected for ChEMBL database [83], our dataset exclusively includes molecules with graph diameters between 17 and 40, clearly ensuring the presence of significant long range dependencies that thoroughly test model capabilities. In the `ECHO-Chem` dataset, each graph represent a single molecule (see Figure 2), and each node (i.e., atom) is labeled with the atomic number, essential for chemical identity, and spatial distance from the center of mass of the molecule, to provide geometrical context. Edges correspond to chemical bonds, and are labeled with bond type (single, double, triple, or aromatic) and bond length. Notably, this encoding of spatial information is invariant under the action of the E(3) group, meaning that relative geometric features such as distances remain invariant under global 3D rotations, reflections and translations of the molecular structure. This ensures that the spatial representation respects the underlying symmetries of molecular physics, essential for learning physically consistent models.

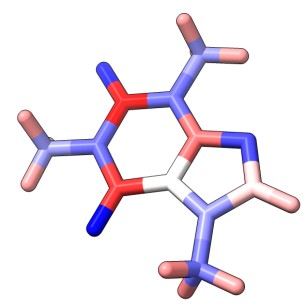

Figure 2: 3D molecular graph of *caffeine* annotated with atomic partial charges. Blue indicates regions of negative partial charge, while red corresponds to positive charge accumulation. Each node is labeled with the atomic number and its distance from the molecule's center of mass, while edges are labeled with bond type and length. The task is to predict the partial charge at each node.

To generate the dataset, we employed a two-steps approach. Firstly, the generation process began with molecular 3D structure generation starting from ChEMBL SMILES [80] strings for all the molecules satisfying the given diameter constraint. In order to generate molecular conformations we opted for coordinates optimization using the Generalized Amber Force Field (GAFF) [37], a well-established force field, specifically designed for optimizing a wide variety of organic and medical interest compounds. These optimized structures, will serve as initialization for the subsequent quantum chemical calculations to determine accurate structures and partial charges. Specifically, we utilized the Hartree-Fock methods with three empirical corrections (`HF-3c`) [75]. The chosen approach balanced computational tractability with the chemical accuracy required for reliable molecular property annotation. All the computations were run thanks to the ORCA package for quantum chemistry [57, 59, 58]. A detailed description of the quantum simulations is provided in Appendix G, along with information about the computing platform in Appendix H.

**Dataset Split.** To evaluate model performance under consistent and reproducible conditions, we employed a random uniform sampling strategy to split the original `ECHO-Chem` dataset. This approach ensures a balanced distribution of molecular structures and charge ranges across the training, validation, and test sets, therefore minimizing potential sampling bias. The dataset was partitioned into 80% for training, 10% for validation, and 10% for testing. This standard 80/10/10 split allows for robust model selection and generalization assessment while preserving the diversity and complexity inherent to the original data.

## 4 Experiments

**Baselines.** We consider a diverse set of GNNs baselines that capture core directions in the development of graph neural architectures, spanning from classical GNNs to more recent approaches that demonstrate strong empirical performance in capturing long-range dependencies. As classical

Table 2: Test MAE (mean with standard deviation as subscript) for each model across the three synthetic tasks: `diam`, `ecc`, and `sssp`. Lower is better. Values are color-coded by performance, with darker green indicating lower error.

| Model | diam ↓ | ecc ↓ | sssp ↓ |
|---|---|---|---|
| GCN | $3.832_{\pm 0.262}$ | $5.233_{\pm 0.034}$ | $2.102_{\pm 0.094}$ |
| GraphCON | $2.969_{\pm 0.189}$ | $5.474_{\pm 0.001}$ | $5.734_{\pm 0.011}$ |
| GPS | $2.160_{\pm 0.098}$ | $4.758_{\pm 0.021}$ | $\mathbf{0.472}_{\pm 0.050}$ |
| GCNII | $2.005_{\pm 0.093}$ | $5.241_{\pm 0.030}$ | $2.128_{\pm 0.429}$ |
| GIN | $1.630_{\pm 0.161}$ | $4.869_{\pm 0.092}$ | $2.234_{\pm 0.271}$ |
| PH-DGN | $1.627_{\pm 0.398}$ | $5.068_{\pm 0.126}$ | $1.323_{\pm 0.485}$ |
| DRew | $1.243_{\pm 0.047}$ | $\mathbf{4.651}_{\pm 0.020}$ | $1.279_{\pm 0.011}$ |
| A-DGN | $1.151_{\pm 0.038}$ | $4.981_{\pm 0.037}$ | $1.176_{\pm 0.140}$ |
| SWAN | $\mathbf{1.121}_{\pm 0.070}$ | $4.840_{\pm 0.045}$ | $0.896_{\pm 0.232}$ |

baseline models, we include GCN [50], GIN [82], GINE[1] [47]and GCNII [13], which represent standard message-passing frameworks with strong theoretical grounding. We also considere multi-hop GNNs, i.e., DRew [39], which adaptively rewire the graph to to facilitate more effective information aggregation across distant nodes. Moreover, we evaluate GPS [64], an effective graph transformer that enables effective long-range propagation via attention mechanism between any pairs of nodes. Finally, we explore the performance of a family of GNNs that draw on principles from dynamical systems theory, namely differential-equation inspired GNNs (DE-GNNs). This includes GraphCON [66], which treats node features as coupled oscillators, as well as models explicitly designed to perform long-range propagation, whose architectures are based on non-dissipative or port-Hamiltoninan dynamics, such as A-DGN [34], SWAN [35], and PH-DGN [44]. Specific configurations of these methods are detailed in Appendix D.

**Model Architecture and hyperparameter selection.** All models share a unified backbone design to enable a fair comparison. In particular, each model is composed of a linear embedding layer, a stack of GNN layers, and a task-specific readout module. For node-level tasks, the readout is a two-layer MLP applied directly to the node representations. For graph-level tasks, node representations are first aggregated using the mean, max, and sum operations, concatenated, and then processed by a two-layer MLP. This standardization ensures that differences in performance are attributable to the core propagation mechanisms rather than auxiliary architectural choices.

Training follows a consistent protocol across all models. We minimize the base-10 logarithm of the Mean Squared Error loss (MSE), $\log_{10}(\text{MSE}(y_{\text{true}} - y_{\text{target}}))$, since the predicted values can be very small in magnitude and this scale-sensitive loss emphasizes small differences. We use the Adam [49] optimizer and adopt Early Stopping based on validation loss. with a patience of 100 epochs. The maximum number of training epochs is set to 1000. This procedure ensures convergence while preventing overfitting, and serves as a reference setup to facilitate reproducibility of our results. In order to ensure a fair and robust comparison across all methods and datasets, we employ an extensive hyperparameter optimization protocol. Specifically, for each model-dataset pair, we perform a Bayesian Optimization based on a Gaussian Process prior [72] in the chosen hyperparameter space, spanning 100 trials to explore the respective search space efficiently. We report the complete set of explored hyperparameters for each model, as well as with the selected hyperparameters, in Appendix D. Finally, the best configuration found is validated through four independent training runs, each initialized with a different random seed. This multi-seed evaluation mitigates the effect of stochastic factors and ensures statistical soundness of the reported results.

**Results on `ECHO-Synth` dataset.** We report results on the synthetic benchmarks in Table 2. All the values are reported using the Mean Absolute Error (MAE). Additional training metrics, particularly MSE and the training loss $\log_{10}(\text{MSE})$, are reported in the Appendix C, Table 4. We clearly observe that models employing global attention mechanisms significantly outperform traditional message-passing frameworks. Specifically, GPS demonstrates superior performance on the `sssp` task, achieving a remarkably low MAE of $0.472$. In line with literature findings [25], this result suggests

---

[1]We added GINE as a baseline to `ECHO-Chem` benchmark to overcome the limitations of GIN to process edge attributes.

that incorporating transformer-like global attention substantially mitigates inherent limitations in localized message-passing, which are pronounced in classic architectures such as GCN and GIN.

Interestingly, it is possible to notice that differential-equation-inspired architectures, particularly those employing non-dissipative or port-Hamiltonian formulations like SWAN, A-DGN, and PH-DGN, consistently perform well across tasks, with similar performance metrics. Notably, SWAN achieves the lowest MAE on the `diam` task (1.121), closely followed by A-DGN and PH-DGN. This highlights the benefit of incorporating non-dissipative dynamics to improve long-range information propagation, thereby preserving critical structural information across extensive message-passing steps. Moreover, the multi-hop GNN, DRew, reveals its effectiveness in the `ecc` task, attaining the lowest MAE (4.651). This success emphasizes the advantage of dynamically rewiring graph structures, thus effectively addressing topological bottlenecks critical for accurately capturing node eccentricities. Differently, GraphCON do not inherently outperform traditional methods, and show notably weaker performance relative to other models of the same architectural family (e.g., A-DGN and SWAN). Thus, mere message-passing dynamics without explicit structural constraints or weight regularization does not ensure superior performance in long-range tasks.

Finally, traditional message-passing models like GCN demonstrate consistent limitations across all benchmarks, indicative of fundamental constraints in purely localized message-passing architectures when facing extensive long-range dependencies as required in our `ECHO-Synth` benchmark suite. This limitation is most evident in the `diam` task, where GCN records the highest MAE (3.832), underscoring its inadequate capacity for global information aggregation.

**Results on `ECHO-Chem` dataset.** We finally detail the performance of all evaluated models on the atomic partial charge prediction task in Table 3. As anticipated, architectures capable of handling long-range dependencies demonstrate a clear advantage, given the nature of the task which requires precise modeling of subtle interatomic interactions spread across the molecular graph. Notably, GPS achieves the best performance across all metrics, with the lowest MAE ($5.65 \times 10^{-3}$) and MSE ($2.00 \times 10^{-4}$), confirming the utility of global attention mechanisms in capturing distant influences that modulate partial charges. This highlights how transformer-style architectures can successfully overcome the locality bottleneck of standard message passing, particularly in chemically meaningful spatial graphs, at the cost of increased computational complexity (as shown in Appendix F).

Models like PH-DGN, A-DGN, and SWAN also yield competitive performance, consistently appearing among the top performers. Their success suggests that imposing non-dissipative priors-such as antis-symmetric weght-space regualation and port-Hamiltonian dynamics-not only regularizes the learning dynamics but also guides the model toward chemically plausible solutions. Indeed, PH-DGN achieves the second-best performance, achieving an MAE of $7.92 \times 10^{-3}$.

The multi-hop GNN, DRew, also achieve strong performance, closely rivaling A-DGN and PH-DGN. Its capacity to adapt the graph structure during training likely enables better long-range signal flow, addressing issues such as topological bottlenecks and poor gradient propagation that are prevalent in molecular graphs. In contrast, GraphCON, which relies on continuous-time dynamics without explicit structural adaptation or attention, fails to deliver comparable performance, achieving one of the worst MAEs ($15.20 \times 10^{-3}$). This reinforces that continuity alone is insufficient for tasks requiring fine-grained long-distance interactions.

Figure 3: Visualization of prediction errors for the `ECHO-Chem` task using two different GNN architectures: GPS Transformer (a) and GCN layer (b). The coloring represents the logarithm of the absolute prediction error, $\log(|y_{\text{true}} - y_{\text{pred}}|)$. Lower values (in green) indicate better prediction accuracy, while higher values (in orange) correspond to larger errors.

Traditional message-passing networks, particularly GCN and GIN, again lag behind, with MAEs exceeding $12 \times 10^{-3}$. These results again confirm the hypothesis that localized aggregation—without mechanisms to integrate distant node information—is inadequate for atomic-level charge modeling. The `ECHO-Chem` benchmark thus clearly illustrates the necessity for architectures

Table 3: Test performance across models on the ECHO-Chem Metrics are reported as mean with standard deviation as subscript. MSE is scaled by $10^{-4}$ and MAE by $10^{-3}$. Lower values are better. Cells are color-coded by performance, with darker green indicating lower error.

| Model | Test Loss | Test MSE ($\times 10^{-4}$) $\downarrow$ | Test MAE ($\times 10^{-3}$) $\downarrow$ |
|---|---|---|---|
| A-DGN | $-3.547_{\pm 0.05}$ | $2.98_{\pm 0.04}$ | $8.47_{\pm 0.05}$ |
| DRew | $-3.532_{\pm 0.03}$ | $3.08_{\pm 0.02}$ | $8.37_{\pm 0.06}$ |
| GCNII | $-3.453_{\pm 0.19}$ | $3.67_{\pm 0.17}$ | $9.26_{\pm 0.14}$ |
| GCN | $-3.136_{\pm 1.40}$ | $6.82_{\pm 2.01}$ | $12.31_{\pm 2.24}$ |
| GIN | $-3.118_{\pm 0.20}$ | $7.76_{\pm 0.36}$ | $13.29_{\pm 0.12}$ |
| GPS | $\mathbf{-3.769}_{\pm \mathbf{0.04}}$ | $\mathbf{2.00}_{\pm \mathbf{0.03}}$ | $\mathbf{5.65}_{\pm \mathbf{0.12}}$ |
| GraphCON | $-3.186_{\pm 0.02}$ | $6.64_{\pm 0.03}$ | $15.20_{\pm 0.05}$ |
| PH-DGN | $-3.604_{\pm 0.02}$ | $2.63_{\pm 0.01}$ | $7.92_{\pm 0.07}$ |
| SWAN | $-3.505_{\pm 0.05}$ | $2.93_{\pm 0.03}$ | $8.79_{\pm 0.06}$ |
| GINE | $-3.481_{\pm 0.23}$ | $3.41_{\pm 0.31}$ | $8.15_{\pm 0.09}$ |

that either incorporate global attention or embed non-dissipative dynamics to effectively tackle the intricate and non-local dependencies inherent in molecular charge distribution.

We provide a visual depiction of charge prediction accuracy on a non-trivial molecule from the test set in Figure 3. The figure contrasts the prediction errors made by two representative GNN architectures: the GPS Transformer (a) and the standard GCN layer (b). Each atom in the molecule is colored according to the logarithm of its absolute prediction error, $\log(|y_{\text{true}} - y_{\text{pred}}|)$, with green tones indicating lower errors and orange tones marking larger discrepancies. As visible in panel (a), GPS yields significantly lower prediction errors across most atomic sites, especially in spatially peripheral regions, reflecting its capacity to capture long-range dependencies and global interactions. In contrast, the GCN model in panel (b) struggles with error accumulation in several areas, particularly at structurally distant or chemically sensitive atoms. This comparison visually underscores the advantage of global attention mechanisms for accurately modeling atomic properties in molecular graphs.

Although partial charges errors are small in absolute magnitude across baselines, even subtle deviations – as stated in [22] – on the order of $10^{-4}\,e$ to $10^{-6}\,e$, can lead to significant downstream effects in molecular modeling and reproducibility of results. Therefore, predictive models must target this level of granularity to produce chemically meaningful outputs.

**Additional Experiments and Analysis.** Additional results and a detailed analysis of baseline performance are provided in Appendix B. We investigate the impact of model depth and graph diameter on test performance across all tasks. Training times are reported in Appendix F. These analyses highlight the ability of different architectures to scale with increasing layer count and to handle long-range dependencies, revealing important differences in robustness and generalization behavior.

# 5 Conclusion

In this paper we propose ECHO, a new benchmark for evaluating long-range information propagation in GNNs. Our benchmark included two main tasks – ECHO-Synth and ECHO-Chem – that target long-range communication in both synthetic and real-world settings. The synthetic tasks are designed to predict algorithmic and long-range-by-design graph properties, while the real-world task focuses on long-range charge distribution in molecules. We provided a detailed analysis to demonstrate that the tasks in ECHO genuinely capture long-range dependencies, and we established strong baselines for each task to provide a comprehensive reference point for future research. We acknowledge some limitations in our current work in Appendix I. Our results highlight the limitations of current GNN architectures when faced with long-range propagation challenges, and we believe that ECHO will serve as a critical step toward building more robust, scalable, and generalizable GNNs capable of handling the full spectrum of graph-based learning tasks, posing a challenge to the community to push the boundaries of GNN design and evaluation.

**Impact Statement.** This work aims to advance the field of machine learning on graphs, focusing on accelerating and advancing the design of more effective GNNs. There are many potential societal consequences of our work, none which we feel must be specifically highlighted here.

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
