# OpenReview forum: "Can You Hear Me Now? A Benchmark for Long-Range Graph Propagation"
_NeurIPS.cc/2025/Datasets_and_Benchmarks_Track — Submitted to NeurIPS 2025 Datasets and Benchmarks Track_

### Official Review · Reviewer_YKmC · 2025-06-28

**Rating:** 4
**Confidence:** 4

**Summary:**

This paper introduces ECHO, a new benchmark designed to evaluate the long-range propagation capabilities of GNNs. ECHO includes three synthetic tasks focused on shortest path prediction and a real-world task for predicting atomic partial charges in molecules. The benchmark systematically assesses how well different GNN architectures capture long-range dependencies on graphs with propagation distances of up to 40 hops.

**Dataset Code Accessibility:**

Yes

**Ethical Considerations:**

No, there are no or only very minor ethics concerns

**Final Justification:**

The concerns that initially led to a borderline reject rating have been addressed.

**Limitations Weaknesses:**

1. While the ECHO benchmark designs meaningful tasks for sssp, ecc, and diam, the graphs used are relatively small, averaging around 100 nodes, which limits its applicability for evaluating long-range dependencies in realistic large-scale settings. It would strengthen the benchmark to include graphs with tens of thousands of nodes.
2. The benchmarking may have limitations. Recent work [1, 2, 3] suggests that classic GNNs (e.g., GCN) may be underestimated on general graph tasks and can achieve state-of-the-art performance with appropriate hyperparameter tuning. To better reflect their true potential, it would be helpful to consider the impact of hyperparameters, even without fully rerunning all settings:
   - Given that the experiments evaluate GNNs with up to 40 layers, it is important to incorporate residual connections to stabilize deep GNN training.
   - It would also be valuable to assess the use of dropout, normalization (BN or LN), FFN, and PE within the GNN baselines, as discussed in [2].
3. The appendix reports the computation time for traditional QM methods per molecule as well as the inference time for GNN models. It would strengthen the paper to highlight this point explicitly in the main text.
4. Given that QM simulation times are often acceptable in many workflows, the practical necessity and impact of using GNN predictions to replace or augment QM pipelines require clearer justification.

[1] Where Did the Gap Go? Reassessing the Long-Range Graph Benchmark, LoG 2023.

[2] Classic GNNs are Strong Baselines: Reassessing GNNs for Node Classification, NeurIPS 2024.

[3] Can Classic GNNs Be Strong Baselines for Graph-level Tasks? Simple Architectures Meet Excellence, ICML 2025.

**Strengths Contributions:**

1. The paper is well-motivated, focusing on a notable gap in current GNN benchmarks that often fail to rigorously test long-range dependency capture.
2. ECHO’s synthetic tasks are well-designed to test long-range propagation, using clear tasks (sssp, ecc, diam) that inherently require global graph information.
3. The ECHO-Chem task is a valuable addition, moving beyond synthetic evaluations by targeting a practical, chemically relevant property prediction task that benefits from long-range interaction modeling.

---

> ### Author Rebuttal · Authors · 2025-07-30
>
> We really thank the reviewer for highlighting the strong motivation of our work in addressing the gap in existing GNN benchmarks for long-range dependency evaluation. We appreciate the reviewer’s recognition of the well-designed synthetic tasks that effectively require global graph information, as well as the value of the ECHO-Chem task for its practical relevance in chemistry. We thank the reviewer again for their review and insightful comments, which we address below. We hope these clarifications and additions address the reviewer’s concerns and that they will consider raising the score.
>
>
> **Update**: in addition to the original real-world node-level task, we add a new molecular energy prediction task, namely ECHO-Energy, where the goal is to predict the total energy of molecules. This task is fundamentally graph-level, as the target property (total energy) depends on the entire molecular structure and cannot be inferred from any subgraph or local neighborhood alone. The results and discussion are included in our response (Table 1) to Reviewer Lfg6.
>
> 1. We thank the reviewer for the insightful suggestion. We would like to point out that our proposed datasets already push beyond the scale of most existing benchmarks in the literature. Increasing graph sizes to tens of thousands of nodes would raise significant scalability challenges and limit accessibility, as only groups with large computational resources (multiple powerful GPUs) would be able to run the benchmark. Moreover, this would raise sustainability concerns due to the high computational cost involved in testing.
> We emphasize that our goal was to provide a benchmark that is practical, accessible, and already extends beyond current standards, striking a balance between long-range complexity and usability by the broader community.
> 2. We thank the reviewer for the valuable suggestions. We would like to clarify that the use of residual connections in GCN was already considered by the  GCNII baseline, but it still does not result in performance to the level of  top models in our benchmark. Therefore, this suggests that residual connection alone is not enough for long-range propagation. Moreover, we adopted the GPS transformer as a baseline, which already includes residual connections and LayerNormalization, and it is one of the top-performing models in our benchmark.
>
>     Regarding the use of dropout, normalization (BatchNorm and LayerNorm), feed-forward networks, and positional encodings, our goal was to evaluate each baseline model as originally proposed in the literature.
>     Nevertheless, we appreciated the reviewer's suggestion, and we have now included additional experiments with GCN, GIN, and GCNII enhanced by PE, BN, and LN as requested. In particular, we modified the original architectures by adding LayerNormalization after each convolution and a residual connection (also in GCNII as an additional, non-gated residual), as well as dropout. We also added LapPE [1] to the node-level features. We report below the results of these additional experiments.
>
>
>     Table 1: Results with Residuals + LayerNorm + Dropout (no LapPE)
>
>     | Model    | diam                    | ecc                     | sssp                    |
>     |----------|-------------------------|-------------------------|-------------------------|
>     | GCNII | $5.024 \pm 0.023$       | $4.929 \pm 0.041$       | $1.272 \pm 0.059$       |
>     | GCN  | $5.833 \pm 0.091$       | $\mathbf{4.838 \pm 0.025}$ | $1.294 \pm  0.098$   |
>     | GIN  | $\mathbf{1.398 \pm 0.025}$ | $4.896 \pm 0.112$     | $\mathbf{1.237\pm0.089}$ |
>
>     Table 2: Results with Residuals + LayerNorm + Dropout + LapPE
>
>     | Model    | diam                | ecc                  | sssp                |
>     |----------|------------------|------------------|------------------|
>     | GCNII | $2.673 \pm 0.031$                |  $2.907 \pm 0.066$                    |  $\mathbf{1.145 \pm 0.072}$    |
>     | GCN  | $4.913 \pm 0.121$                 |  $\mathbf{2.091 \pm 0.098}$     |  $1.221 \pm 0.103$             |
>     | GIN  | $\mathbf{1.931 \pm 0.049}$    |  $4.787 \pm 0.137$                    |  $1.611 \pm 0.170$             |
>
>     The results show that while non-long-range-specific models (e.g. GCN) benefit from these additions and achieve improved performance, they still fall short compared to models explicitly designed for long-range propagation, such as SWAN and ADGN.
>  Regarding the use of LapPE, we deliberately chose not to include it in the original benchmark as it contains global information about the graph structure, therefore serving as a possible shortcut for long-range propagation assessment. However, these additional results are valuable and represent a useful extension of our benchmark.
>
> We would appreciate the reviewer’s feedback on whether these extended results, focusing on upgraded versions of standard GNNs, should be included in the main body of the paper or as an appendix, considering the aim of the paper. We are happy to integrate them wherever they would be most helpful to the reader and the community.
>
> 3. We sincerely thank the reviewer for their support in improving our work. We have now highlighted the comparison between traditional QM methods’ computation time and GNN inference time in the main text.
> 4. This is an important point. While it's true that QM simulation times can be acceptable in certain workflows, they often become a significant bottleneck in large-scale applications—for example in molecular dynamics. QM methods like DFT can take from several minutes to hours per molecule, depending on system size and level of theory, making them impractical for use at scale.
> In contrast, GNNs can make accurate predictions in milliseconds, offering orders-of-magnitude speedups. This difference enables applications that would otherwise be computationally infeasible, such as exploring vast chemical spaces or running long-timescale simulations. We have now clarified this motivation more explicitly in the main text.
> ----------------------------
> In light of these clarifications and the additional experiments, we believe our paper provides an effective and well-motivated benchmark, offering a diverse set of tasks and a leaderboard that the community can use to evaluate and develop models targeting improved long-range propagation in graphs. We appreciate the reviewer’s time and hope this addresses the concerns and we hope the reviewer can consider revising their score.
>
> [1] Belkin, Mikhail, and Partha Niyogi. "Laplacian eigenmaps for dimensionality reduction and data representation." Neural computation 15.6 (2003): 1373-1396.

---

> > ### Comment · Reviewer_YKmC · 2025-08-05
> >
> > Thank you for your detailed rebuttal.
> >
> > I appreciate the additional experiments. As you are aware, the results of benchmark baselines are critical. Subsequent works often directly compare against them without re-tuning, and understated baselines can unintentionally exaggerate the effectiveness of new models. For this reason, I recommend including the extended results in the main body of the paper. Establishing well-tuned baselines is essential not only for the strength of this work but also for the broader research community.
> >
> > Given your clarifications, I have raised my score. However, I still strongly recommend considering further tuning for standard baselines, by including MLP/FFN after messag passing and alternative positional encodings beyond LapPE (like RWSE).

---

> > > ### Author Response · Authors · 2025-08-05
> > >
> > > We sincerely thank the reviewer for their understanding and for raising the score. We agree that well-tuned baselines are important for both our work and the broader community.
> > > As we write, we are running additional experiments with MLP/FFN layers after message passing and alternative positional encodings such as RWSE, as suggested.
> > >
> > > These results will be included in the final version of the paper and reported during the discussion if completed in time.
> > >
> > > Thank you again for your valuable support.

---

### Official Review · Reviewer_7v5E · 2025-06-29

**Rating:** 4
**Confidence:** 5

**Summary:**

The paper introduces ECHO, a new benchmark for testing how well graph neural networks pass information over many hops. It has two parts. The first, ECHO-Synth, contains 10,080 synthetic graphs built in six shapes. The second part, ECHO-Chem, is a real dataset of about 196 k molecules. Each molecule is a graph where nodes are atoms and edges are bonds. The authors run nine well-known GNN families. Results show big gaps: models with global attention or special non-dissipative dynamics beat classic GCN and GIN by wide margins on both the synthetic and chemical tasks. These findings confirm that current architectures still struggle when information must travel far.

**Dataset Code Accessibility:**

Yes

**Ethical Considerations:**

No, there are no or only very minor ethics concerns

**Final Justification:**

Thanks for the authors' rebuttal. My concerns have been well addressed. Therefore, I maintain my original score.

**Limitations Weaknesses:**

1. Real world datasets rarely start from feature-free nodes, while success on purely structural tasks may not translate once rich, noisy attributes interact with long paths.
2. For ECHO-Chem,  the baseline pool omits modern 3-D equivariant models such as GemNet [1] and MACE [2].
3. JK-net[3] and NDLS[4] (e.g., Eq.(3) in NDLS)  both quantify long-range influence by examining how a small perturbation to a neighbor’s input features changes the center node’s representation after multiple message-passing steps.  Such a influence measure would make the benchmark a stronger litmus test for true long-range reasoning, aligning the measured receptive field with the benchmark’s 17-40 hop span.

[1] GemNet: Universal Directional Graph Neural Networks for Molecules.

[2] MACE: Higher Order Equivariant Message Passing Neural Networks for Fast and Accurate Force Fields.

[3] Representation Learning on Graphs with Jumping Knowledge Networks.

[4] Node Dependent Local Smoothing for Scalable Graph Learning.

**Strengths Contributions:**

1. Every task in ECHO is built so that useful signals must travel exactly 17 – 40 hops across the graph. The paper states this range for both the synthetic suite and the 196 Chem dataset. This explicit target distance is rare in earlier benchmarks. I think the controllable propagation ranges and the pre-defined optimal propagation hops are very important for evaluating long-range GNNs.
2. The three synthetic tasks are reasonable, because all require traversing the entire graph, and they cannot be solved by counting local patterns.
3. Tables 2 and 3 show large performance gaps: GPS and non-dissipative ODE-style networks sharply beat GCN/GIN on both synthetic and chemical tasks, underscoring why new methods are still needed.

---

> ### Author Rebuttal · Authors · 2025-07-30
>
> We thank the reviewer for highlighting the importance of our design choice to enforce signal propagation across long hops in both the synthetic and molecular tasks. We appreciate the reviewer's recognition of the value of controllable propagation ranges and explicitly defined optimal distances, which we believe are crucial for meaningful evaluation of long-range GNNs. We're also glad the reviewer found the synthetic tasks well-motivated and appreciated the significant performance differences in our benchmarks, which point to the continued need for improved architectures.
>
> **Update**: in addition to the original real-world node-level task, we add a new molecular energy prediction task, namely ECHO-Energy, where the goal is to predict the total energy of molecules. This task is fundamentally graph-level, as the target property (total energy) depends on the entire molecular structure and cannot be inferred from any subgraph or local neighborhood alone. The results and discussion are included in our response (Table 1) to Reviewer Lfg6.
>
> We thank the reviewer again for the review and thoughtful comments, which we address below.
>
> 1. We would like to clarify that our real-world dataset, ECHO-Chem, does include rich node and edge features as discussed in Sec 3.2. Specifically, node features consist of atomic number and distance from the molecule’s center of mass, while edge features include bond type and bond length. We revised this discussion in the main text to avoid any further confusion.
> Additionally, even in our synthetic single-source shortest path task, nodes are not feature-free: each node includes a binary feature indicating whether it is the source node. This ensures that the model utilizes meaningful input features beyond just the structure, and this information is essential for solving the task.
> 2. We understand the reviewer’s perspective and its relevance when considering molecular tasks. However, the primary goal of our selection of models is that of providing a consistent leaderboard of widely acknowledged baselines from diverse and general GNN families. Including 3-D equivariant models would not be meaningful for our synthetic tasks, which do not involve 3-D geometric information, resulting in different leaderboards for the different benchmarks and possibly raising questions as regards why those specific 3-D equivariant models were chosen over others, which may offer solid grounds for criticisms by the community. We therefore would like to remain with our choice of general GNN models, leaving the assessment and proposal of 3-D equivariant models as future work from the community contributors.
>
> 3. We acknowledge the reviewer’s thoughtful input, and we included JK-net as a representative baseline in our experiments, as it further strengthens the benchmark’s evaluation of long-range propagation. For convenience, we report our results in the table below, on the ECHO-Synth benchmark:
>
> Table 1.
> | Model    | diam   | ecc    | sssp   |
> |----------|-----|-----|-----|
> | JK-net + GCN  | $7.099 \pm 0.269$ | $5.104 \pm 0.011$ | $1.468 \pm 0.074$ |
> | JK-net + GCNII | $3.846 \pm  0.151$ | $5.456 \pm 0.012$ | $2.449 \pm 0.031$|
> | JK-net + GIN  | $9.020 \pm  0.051$ | $4.943 \pm 0.103$ | $1.177 \pm 0.439$ |
>
> We will include these results in the updated version of the paper. We are also in the process of  running JK-net on the ECHO-Chem and ECHO-Energy tasks.

---

### Official Review · Reviewer_Lfg6 · 2025-07-03

**Rating:** 4
**Confidence:** 3

**Summary:**

This paper introduces ECHO, a benchmark suite for evaluating long-range information propagation in Graph Neural Networks (GNNs). It includes three synthetic tasks across diverse graph topologies and a novel real-world dataset, ECHO-Chem, focused on atomic charge prediction in molecules. The benchmark highlights key challenges in long-range reasoning and is supported by thorough empirical analysis, strong baselines, and publicly available code and data.

**Additional Feedback:**

1. Consider including a classification task (e.g., graph property prediction) in the synthetic benchmarks to broaden applicability.

2. If possible, include visualizations of learned attention patterns or message paths to illustrate how long-range information is (or is not) propagated.

3. Investigate robustness by randomly dropping edges or adding noise to node features. This would clarify which models are most fragile under signal degradation.

**Dataset Code Accessibility:**

Yes

**Dataset Code Comments:**

The datasets and code can be available at the provided links.

**Ethical Considerations:**

No, there are no or only very minor ethics concerns

**Final Justification:**

Most of my concerns have been addressed and I maintain my positive assessment of the paper.

**Limitations Weaknesses:**

1. The benchmark is currently limited to regression and algorithmic prediction tasks; expanding to include classification or downstream application tasks could increase its utility.

2. While the synthetic tasks are diverse, the paper could benefit from a deeper discussion on why certain architectures perform better under specific topological conditions.

3. As mentioned in the appendix, ECHO-Chem is computationally expensive to replicate (due to quantum chemistry simulations), which may limit practical reproducibility of dataset generation.

**Strengths Contributions:**

1. The synthetic graphs span a variety of controlled topologies and graph diameters, with clear motivation and task definitions.

2. The paper provides thorough benchmarking across a diverse set of GNN architectures, including multi-hop, transformer-based, and physics-informed models.

3. The authors systematically evaluate performance across six graph topologies (line, ladder, grid, tree, caterpillar, lobster) and a range of graph diameters (17– 40 hops) for all three synthetic tasks.

4. The dataset and code are openly available, and experimental protocols (including hyperparameter tuning) are detailed.

---

> ### Author Rebuttal · Authors · 2025-07-30
>
> We thank the reviewer for recognizing the diversity and motivation behind our synthetic graph topologies and tasks, as well as the systematic evaluation across different diameters and structures. We appreciate the reviewer’s positive feedback on the range of GNN architectures considered, the transparency of our experimental protocol, and the availability of our dataset and code.
>
> 1. We agree that a broadening of the scope of the benchmark in terms of the variety of learning tasks being tackled would further strengthen the impact of our work. That is why we are now introducing an additional real-world task, this time exercising predictions at a graph level rather than at a node level. Specifically, we add a new molecular energy prediction task, namely ECHO-Energy, where the goal is to predict the total energy of molecules. This task is fundamentally graph-level, as the target property (total energy) depends on the entire molecular structure and cannot be inferred from any subgraph or local neighborhood alone.
> Accurately estimating molecular energy requires capturing global interactions between atoms across the whole molecule, making it a suitable testbed for evaluating long-range message-passing capabilities in GNNs.
> This task is particularly relevant in the context of quantum chemistry, where predicting molecular properties is crucial for understanding chemical behavior and reactivity. The dataset consists of approximately 155k molecules with varying sizes and structures, providing a rich set of examples for evaluating GNN performance on long-range dependencies. We used the same experimental protocol adopted for the ECHO-Chem task, including model selection, training setup, and hyperparameter tuning. Moreover, the molecular graphs are annotated with DFT-level atomic charges, which provide rich node-level signals grounded in quantum chemical computations.
> In the following table (Table 1), we report results for this new task and compare the performance of various architectures under consistent evaluation settings (both test MAE and test loss as defined in the main text).
>
>    Table 1.
>    | Model       | Test MAE | Test Loss |
>    |-------------|-------------|--------------|
>    | DRew   | $15.066 \pm 8.379$    | $-4.712 \pm 0.0247$   |
>    | GCN     | $38.775 \pm 16.317$   | $-3.9775\pm 0.363$   |
>    | **GPS** | $\mathbf{4.777 \pm 0.189}$ | $\mathbf{-5.859 \pm 0.011}$ |
>    | ADGN          | $12.407\pm 4.285$   | $-4.845 \pm 0.021$   |
>    | GCNII          | $12.042 \pm 2.565$   | $-4.999 \pm 0.146$   |
>    | SWAN         | $12.902 \pm 3.399$   | $-4.908 \pm 0.416$    |
>    | GIN              | $34.782\pm 3.962$   | $-4.078 \pm 0.159$   |
>    | GINE           | $18.551\pm 3.679$   | $-4.615\pm 0.114$  |
>    | PHDGN       | $15.483 \pm 0.621 $   | $-4.741 \pm 0.024$   |
>
>     Again, the results on the molecular energy prediction task reveal clear differences in the ability of GNN models to capture long-range dependencies and global molecular structure. GPS achieves the best overall performance, with the lowest test MAE (4.777) and test loss, highlighting the effectiveness of attention-based message passing in modeling complex molecular interactions. Non-dissipative architectures perform similarly and achieve good overall performance.
>
> 2. We thank the reviewer for their valuable comment. We agree that further discussion on the interaction between model design and graph topology can enhance the paper. In general, the graph topologies employed in our benchmark are characterized by structural bottlenecks, i.e., edges with high negative curvature [1]. Such bottlenecks make long-range propagation more difficult, as they concentrate exponentially growing messages into a single node. In this scenario, architectures such as transformers or methods like DRew (which incorporate rewiring mechanisms) can alleviate this issue by bypassing these structural constraints.
> On the other hand, non-dissipative message passing models tend to preserve signal strength better within the graph, improving general propagation. However, even these methods cannot fully overcome the limitations imposed by the graph topology, and their performance remains partially affected by the presence of structural bottlenecks. We added this discussion in the revised version of the paper.
>
>     [1] Topping et al. Understanding over-squashing and bottlenecks on graphs via curvature. ICLR 2022
> 3. While it is true that reproducing ECHO-Chem from scratch would require significant computational resources due to the quantum chemistry simulations, we emphasize that this is not necessary, as we are releasing the full dataset to ensure full accessibility and reproducibility of our experiments.
> Moreover, the computational cost of generating the dataset highlights the real-world relevance of this task. GNNs can approximate these expensive quantum chemical properties in a matter of seconds, offering substantial speed-ups that are highly valuable in practice.
> We are also mindful of sustainability and accessibility. In designing ECHO, we have prioritized tasks that are both impactful and usable without requiring high-end infrastructure, avoiding digital divide concerns while still reflecting real-world scientific challenges, particularly those where running the original experiments would be prohibitively costly.
>
> ----
> As regards the additional feedback:
> 1. As mentioned above, we agree that extending the benchmark with classification tasks is a valuable direction. While our current focus is on regression and algorithmic prediction tasks, we have included a novel graph-level regression task (molecular energy prediction) to broaden the scope, and we plan to explore classification tasks in future work.
>
> 2. We thank the reviewer for this excellent and valuable suggestion. We fully agree that visualizing learned attention patterns can provide valuable insight into how long-range information is propagated, particularly across different topologies. We have explored this direction and found the results to be very informative. In particular, observed attention scores revealed interesting learned message-passing patterns, which are specific for each topology.  For instance, in the lobster topology, nodes directly connected to the central backbone often act as intermediaries for message exchange between distant parts of the graph. This behavior is reflected in higher attention coefficients assigned to these nodes, effectively revealing how the model routes long-range communication through structural shortcuts, thus confirming the long-range nature of the proposed tasks.
> While we are unable to include figures in the rebuttal, due to the rules enforced aposteriori by the PC, we plan to add these visualizations in the updated version of the paper. These will include both attention heatmaps and graph renderings with overlaid edges indicating strong attention weights, to better illustrate the propagation dynamics.
>
> 3. This is an interesting suggestion, but  while investigating robustness under perturbations is certainly valuable, it falls outside the primary scope of our work. Our goal is to introduce a benchmark specifically designed to evaluate long-range information propagation in GNNs, a capability for which solid, targeted benchmarks are currently lacking. With this in mind, our focus is on providing a well-motivated and diverse set of tasks, along with a leaderboard that the community can use to assess and develop models aimed at improving long-range communication. Evaluating model robustness under noise is an interesting orthogonal direction, which we leave for future work.

---

> > ### Comment · Reviewer_Lfg6 · 2025-08-04
> >
> > Thank you for your detailed response and my concerns have been addressed. I maintain my positive assessment of the paper.

---

> > > ### Author Response · Authors · 2025-08-04
> > >
> > > We thank the reviewer for their thoughtful review and engagement, and we hope that our clarifications have further strengthened your evaluation toward full acceptance of our work. Nevertheless, we remain available to provide any additional clarification during the discussion phase, if needed.

---

### Official Review · Reviewer_wDaK · 2025-07-05

**Rating:** 4
**Confidence:** 3

**Summary:**

The paper introduces ECHO, a novel benchmark designed to evaluate the capability of GNNs to propagate information over long ranges.
It features two main tasks (i.e., synthetic ECHO-Synth dataset: proposed topologies like line, ladder, grid, tree, caterpillar, and lobster; real-world ECHO-Chem dataset: molecular charge prediction), which focus on capturing long-distance dependencies.
The authors conduct experiments on these two datasets with widely used GNN and Graph Transformer frameworks. Results show that models with global attention, such as GPS Transformer, outperform standard GCNs in capturing long-range dependencies, especially in complex molecules.

**Dataset Code Accessibility:**

Yes

**Dataset Code Comments:**

The datasets and code are publicly released, well-documented, and in a final, reproducible format, with detailed instructions.

**Ethical Considerations:**

No, there are no or only very minor ethics concerns

**Limitations Weaknesses:**

1. While the benchmark covers a solid range of GNNs, it mostly sticks to established families (e.g., message-passing, attention-based, DE-GNNs), with limited exploration of adaptive methods like DeeperGCN [1] and DAGNN [2].
```
[1] Li, Guohao, et al. "Deepergcn: All you need to train deeper gcns." arXiv preprint arXiv:2006.07739 (2020).

[2] Liu, Meng, Hongyang Gao, and Shuiwang Ji. "Towards deeper graph neural networks." Proceedings of the 26th ACM SIGKDD international conference on knowledge discovery & data mining. 2020.
```
2. It would be helpful to include a GCN performance comparison on traditional graph and molecule datasets to better highlight the unique challenges of long-range tasks.

**Strengths Contributions:**

1. Traditional GNNs often struggle with over-smoothing and capturing long-range dependencies. ECHO directly tackles this by focusing on evaluating long-range communication through a mix of synthetic and real-world tasks.

2. The paper provides a clear and well-structured way to measure how well different GNNs handle non-local interactions—something that’s been hard to pin down in past benchmarks.

3. The benchmark sets up strong baselines and does a good job highlighting where current methods fall short (e.g., comparing GCN with GPS Transformer), which can help drive progress in building more powerful and scalable GNNs, especially in fields like chemistry and network science.

4. The insights about global attention being useful for handling long-range dependencies are interesting and feel intuitive.

---

> ### Author Rebuttal · Authors · 2025-07-30
>
> We thank the reviewer for acknowledging the novelty of our benchmark as well as our effort to address long-range propagation in GNNs through a mix of synthetic and real-world tasks. We appreciate the reviewer’s positive feedback on the clarity and effectiveness of our benchmark for evaluating non-local node communication, while setting up strong baselines for highlighting current GNNs' limitations.
>
> **Update**: in addition to the original real-world node-level task, we add a new molecular energy prediction task, namely ECHO-Energy, where the goal is to predict the total energy of molecules. This task is fundamentally graph-level, as the target property (total energy) depends on the entire molecular structure and cannot be inferred from any subgraph or local neighborhood alone. The results and discussion are included in our response (Table 1) to Reviewer Lfg6.
>
> ---
>
> 1. Insightful suggestion: we are including additional baselines in the updated version of the paper. We note that our current selection focuses on architectures that are widely recognized as reference baselines in the literature on long-range propagation in GNNs. This choice was made to ensure a clear and fair comparison across established approaches. Motivated by this, we have added DAGNN as a valuable baseline and representative adaptive model to our benchmark. To ensure a fair and consistent evaluation, we ran DAGNN using its original source code on our synthetic tasks. For each task, we performed 50 steps of Bayesian optimization to tune hyperparameters, following the same protocol applied to other baselines in our benchmark. This setup allows for a reliable comparison under equal search budgets and training conditions. We are also in the process of extending the results for the new baseline for the ECHO-Chem and ECHO-Energy datasets. The table below presents DAGNN performance on the ECHO-Synth task.
>     | Model    | diam | ecc | sssp |
>     |----------|------|-----|------|
>     | DAGNN    | $5.112\pm 0.026$ | $5.554 \pm 0.001$  | $6.108\pm0.003$  |
> 2. We thank the reviewer for the suggestion, however, rather than attracting too much attention on a single model (e.g. GCN) and its limitations, we would like to focus on the specific problem of long-range propagation while keeping an agnostic view on models. That is why we would like to avoid indulging too much in highlighting aspects of the limited performance of GCN outside what is being shown in the benchmark.

---

### Decision · Program_Chairs · 2025-09-18

**Decision:**

Reject

**Comment:**

This submission presents ECHO, a benchmark aimed at long-range information propagation in GNNs, and all four reviewers acknowledged its novelty and potential community value. However, several substantive concerns remain unresolved. Reviewer YKmC emphasized that the graphs are relatively small and the classic baselines are insufficiently tuned; while the authors added experiments, the coverage is still incomplete. Reviewer 7v5E highlighted the absence of modern 3D-equivariant models, which limits the chemical relevance of ECHO-Chem. Reviewer Lfg6 recommended expanding the scope beyond regression tasks and evaluating robustness, but these directions were left to future work. Reviewer wDaK also pointed to gaps in baseline diversity. Overall, while the benchmark is promising and could evolve into a valuable community resource, the current version relies on small synthetic settings, introduces late additions that are not fully integrated, and has unresolved issues related to dataset compliance. I therefore recommend rejection, but encourage the authors to resubmit once the above criticism has been addressed.